

# ACO-Kinematic: a hybrid first off the starting block

Kaylash Chaudhary, Avinesh Prasad, Vishal Chand and Bibhya Sharma

The University of the South Pacific, Suva, Fiji

## ABSTRACT

The use of robots in carrying out various tasks is popular in many industries. In order to carry out a task, a robot has to move from one location to another using shorter, safer and smoother route. For movement, a robot has to know its destination, its previous location, a plan on the path it should take, a method for moving to the new location and a good understanding of its environment. Ultimately, the movement of the robot depends on motion planning and control algorithm. This paper considers a novel solution to the robot navigation problem by proposing a new hybrid algorithm. The hybrid algorithm is designed by combining the ant colony optimization algorithm and kinematic equations of the robot. The planning phase in the algorithm will find a route to the next step which is collision free and the control phase will move the robot to this new step. Ant colony optimization is used to plan a step for a robot and kinematic equations to control and move the robot to a location. By planning and controlling different steps, the hybrid algorithm will enable a robot to reach its destination. The proposed algorithm will be applied to multiple point-mass robot navigation in a multiple obstacle and line segment cluttered environment. In this paper, we are considering a priori known environments with static obstacles. The proposed motion planning and control algorithm is applied to the tractor-trailer robotic system. The results show a collision and obstacle free navigation to the target. This paper also measures the performance of the proposed algorithm using path length and convergence time, comparing it to a classical motion planning and control algorithm, Lyapunov based control scheme (LbCS). The results show that the proposed algorithm performs significantly better than LbCS including the avoidance of local minima.

# INTRODUCTION

Over the years, the use of robots has evolved due to improved capacities and abilities to carry out complex and diverse activities in areas such as manufacturing, logistics, home, travel, health, mining, civil, military, and transportation (*Dunbabin & Marques, 2012*; *Khurshid & Bing-rong, 2004*; *Murphy et al., 2009*; *Napper & Seaman, 1989*; *Raj et al., 2018*). In almost all cases, a robot must travel from one point to another in order to complete a task. A robot should also avoid collisions and dangerous situations while navigating through its surroundings in order to reach a certain point. This is generally known as findpath or robot navigation problem which invariably has four categories: localization, path planning, motion control, and cognitive mapping (*Buniyamin et al., 2011*). This paper

Corresponding author
Kaylash Chaudhary,
kaylash.chaudhary@usp.ac.fj

focuses on the motion planning and control problem. Due to its usefulness in real-world applications, robot motion planning and control has been a widely researched topic over the past four decades. The primary goal of robot path planning and control is to identify the most efficient and safe route from point A to point B and subsequently control the robot to point B. There are two subtasks in robot motion planning and control: (1) plan a path which should be obstacle and collision free, and (2) control the robot to its destination.

Various methods are available in the literature for path planning which can be categorized into classical, heuristic and machine and deep learning. Artificial potential field (*Lee & Park, 2003*), cell decomposition (*Kloetzer, Mahulea & Gonzalez, 2015*), road map (*Lingelbach, 2004*), and virtual force field (*Bortoff, 2000*) are examples of classical or traditional approach. Optimization algorithms such as firefly (*Patle et al., 2018*), ant-colony (*Brand et al., 2010*), and particle swarm (*Wang & Zhou, 2019*) are examples of heuristic approaches. Algorithms such as neural networks, decision trees, Nave Baiyes, and others are used in machine and deep learning techniques for robot path planning (*Otte, 2015*). This paper will focus on a heuristic approach, in particular, the ant colony optimization. An Ant colony optimization (ACO) is a popular algorithm that has been applied to different problems, including path planning problems in robotics (*Brand et al., 2010*; *Reshamwala & Vinchurkar, 2013*). In this paper, given the nature of the problem, the authors use the ACO algorithm developed by Socha and Dorigo (*Socha & Dorigo, 2008a*; *Socha & Dorigo, 2008b*) for continuous domain (ACOR); however, there are different variants of ACO algorithms such as fuzzy heuristic based ACO (*Shokouhifar, 2021*), continuous ACO (CACO) (*Bilchev & Parmee, 1995*) and continuous interacting ant colony (CIAC) (*Dréo & Siarry, 2004*) which have been utilized for different problems and situations. The authors chose the variant of ACO proposed by Socha and Dorigo because it clearly outperformed other continuous ACO variants like CACO, API and CIAC in eight test functions (*Socha & Dorigo, 2008a*; *Socha & Dorigo, 2008b*) and it follows the original ACO formulation. Furthermore, Socha and Dorigo also compared ACOR with continuous genetic algorithm (CGA), enhanced continuous tabu search (ECTS), enhanced simulated annealing (ESA) and differential evolution (DE), where ACOR performed well in one third of the test problems and performed not much worse on other problems (*Socha & Dorigo, 2008a*; *Socha & Dorigo, 2008b*).

Once a path is planned the robot can track or follow the planned path to reach its destination. There are different methods in the literature for robot motion control such as kinematics, dynamics, artificial potential field, fuzzy logic and many more (*Martınez-Alfaro & Gomez-Garcıa, 1998*; *Tanaka et al., 1998*; *Tsai & Wang, 2005*; *Watanabe et al., 1998*). This paper will focus on kinematic equations of the robots for motion control.

Majority of the studies in the literature carry out motion control in two different phases: (1) path planning, and (2) subsequent robot tracking or control. This paper will address the problem of the motion planning and control differently. A robot's next step will be planned and the robot will move to that step. At the time of the robot's step planning, the obstacles and other robots will be considered and a collision free step will be determined. By repeating this process, the robot will reach its destination . Therefore, there is path planning and control at every step to its destination.

This research will introduce a new hybrid algorithm that is a strategic combination of an ant colony optimization algorithm and kinematic equations of robot motion. The purpose of the algorithm is to plan a step (location) for the robot and the robot will use its kinematic equations to move to that step (location). This process is repeated until the final destination is reached. This is the main difference when compared to other algorithms and hybrids as most plan the entire path first and then the robot starts the journey. Selected scenarios will be used to showcase the new hybrid and a performance comparison in terms of path length and convergence time will be made with the Lyapunov based Control Scheme (LbCS), a classical approach for motion control.

The main contributions of this paper are as follows:

1. ACO-Kinematic algorithm: A new hybrid algorithm is proposed which plans a step for the robot and the robot moves to that step and this process continues until the robot reaches its destination. In literature, according to authors knowledge, there is no hybrid algorithm that plans next step location of robot and controls it to that step location. The proposed algorithm also solves the problem of local minima.

2. Multi-objective problem formulation: A new multi-objective problem is formulated in terms of path length and safety of the path. The safety objective is achieved through ant colony optimization and the path length objective is achieved by both, ant colony optimization and kinematic equations.

3. Application: The methodology derived for point-mass robots has been successfully applied to tractor-trailer robotic system to show the effectiveness of the proposed algorithm in real life applications.

4. Analysis: The performance of the new hybrid algorithm has been compared to that of lyapunov based control scheme (LbCS) in terms of path length and convergence time. Such a comparison using the LbCS has been carried out for the first time. The analysis show that ACO-Kinematic performed slightly better than LbCS including the avoidance of local minima.

'Related Work' presents the literature review on motion planning and control algorithms. The problem statement is discussed in 'Problem Statement and Objectives'. 'Ant Colony Optimization' discusses the ant colony optimization algorithm. The three objectives of motion planning namely, short, safe and smooth path are formulated and discussed in 'Robot Motion Planning and Control Formulation'. The proposed algorithm is presented in 'Proposed Algorithm'. The three case studies and example scenarios, including the kinematic equations for the tractor-trailer robot are discussed in 'Results'. 'Discussion' presents three different scenarios to measure performance of the proposed algorithm and the LbCS. Finally, the paper concludes in 'Conclusion and Future Work', discussing its contributions and recommendations for future work.

## RELATED WORK

In literature, there are mostly instances of path planning and motion control being researched and implemented separately. There are a few algorithms that consider motion planning and control in parallel or simultaneously. The section will outline these algorithms and present a brief comparison with the proposed algorithm.

Firstly, there are algorithms that carry out path planning and then control the robots on these paths, usually known as path-tracking. For example, Saputro et al. implemented trajectory planning and tracking of mobile robots using a map and predictive control (_Saputro, Rusmin & Rochman, 2018_). A map of the environment is first created and then the collision free path is searched using A* algorithm. Then a model predictive control is used to track the robot on the reference path. Likewise, Ning et al. implemented a trajectory planning and tracking control scheme for autonomous obstacle avoidance of wheeled inverted pendulum (WIP) vehicles (_Ning et al., 2020_). Motion planning and trajectory control has been applied to car parking as well (_Zips, Bck & Kugi, 2013_), where the authors first find a path by solving a static optimization problem and then use a optimal controller for parking. The reader is referred to _Chen, Peng & Grizzle (2017)_, _Guo et al. (2018)_, _Viana et al. (2021)_, _Wang et al. (2020)_ and _Zhang et al. (2019)_ for other examples of such algorithms.

Secondly, there are algorithms that consider path planning and motion control in parallel. _Wahid et al. (2017)_ used artificial potential field method for motion planning and kinematics for the control to design vehicle collision avoidance assistance systems. Lyapunov based control scheme (LbCS) was developed by _Sharma, Vanualailai & Singh (2014)_ and _Sharma et al. (2008)_ and has been used by many researchers in different scenarios for robot motion planning and control (_Devi et al., 2017_; _Kumar et al., 2021_; _Prasad, Sharma & Vanualailai, 2017_; _Prasad et al., 2021_; _Raj et al., 2021_; _Raj et al., 2018_). LbCS is a time-invariant nonlinear method that is used to create velocity or acceleration-based controls enabling a robot to move safely in the workspace while avoiding obstacles. Raj et al. used LbCS to control a system of 1-trailer robots in a cluttered environment including a swarm of boids (_Raj et al., 2021_) and navigate car-like robots in 3-dimensional space (_Raj et al., 2018_). Prasad et al. used LbCS to derive the acceleration-based controllers for the mobile manipulator in 3-dimensional (_Prasad et al., 2021_) and to motion control a pair of cylindrical manipulators in a constrained 3-dimensional workspace (_Prasad, Sharma & Vanualailai, 2017_). Researchers also used LbCS for controlling quadrotors in different environments (_Raj, Raghuwaiya & Vanualailai, 2020_; _Raj, Raghuwaiya & Vanualailai, 2020_; _Vanualailai, Raj & Raghuwaiya, 2021_).

All of the motion planning and control algorithms outlined in the previous paragraph except LbCS first plan the path for the entire journey and then control the motion of the robot on the planned path. However, the proposed algorithm is different from these algorithms as it plans one step at a time and controls the motion of the robot to that step. LbCS also plans a step and then controls the motion of the robot to that step. Therefore, the proposed algorithm will be compared for performance with the LbCS.

## PROBLEM STATEMENT AND OBJECTIVES

Suppose there is an initial position and a target position for each robot in a bounded workspace with several static obstacles of different shapes and sizes. The mobile robots will start at the initial position, avoid obstacles in path and reach their designated target position. Therefore, the main objective of the mobile robots is to reach their target by taking

a shorter and safer route (avoid collision with obstacles and other robots).The following assumptions are made to achieve the objective of motion control in this paper:

**Assumption 1**: The obstacles are of circular and rectangular shapes. In some cases, it is also represented as line segments. These obstacles are of different sizes and randomly distributed in the bounded workspace. The obstacles are static with known locations.

**Assumption 2**: The kinematic equations are used for the motion of point-mass and tractor-trailer robots. The robots can move in any direction.

Since this paper introduces first of a kind hybrid, more features such as handing uncertainties in the environment and locations of the static obstacles and the effects of noise will be added in the future work.

The research objectives of this paper are as follows:

- Design and implement a hybrid algorithm composed of a heuristic and a classical method for planning a step location of robot and controlling it to that step, and hence reach a target with a series of such steps.
- Apply the hybrid algorithm to a real life application such as tractor-trailer robots.

## ANT COLONY OPTIMIZATION

Ant colony optimization (ACO) was initially developed by Dorigo et al. (*Socha & Dorigo, 2008a*; *Socha & Dorigo, 2008b*) in the early 90′s for combinatorial optimization but later modified for continuous domains (*Dréo & Siarry, 2004*; *Socha & Dorigo, 2008a*; *Socha & Dorigo, 2008b*). ACO has been inspired from natural ants, which move out of their nest in search of food and move randomly in the surrounding area. Once an ant finds food, it evaluates and carries it on its back. On the way back to the nest, the ant deposits a pheromone trail on the ground which depends on the amount of food and quality. This pheromone trail guides other ants to the food source (*Socha & Dorigo, 2008a*; *Socha & Dorigo, 2008b*). In this paper, we have used the ant colony optimization for continuous domain (ACOR) proposed by Socha and Dorigo (*Socha & Dorigo, 2008a*; *Socha & Dorigo, 2008b*). The original ACO is based on discrete domain where pheromone and heuristic information are used to make different probabilistic choices. The main idea of ACOR is shifting from a discrete probability distribution to a continuous one that is, a probability density function (PDF). In ACOR, an ant samples a PDF instead of choosing a solution component when compared to ACO in discrete domain. ACOR closely follows the metaheuristic of ACO in discrete domain.

ACO algorithm for continuous domain has two types of population: archive and new. The pheromone information is stored as a solution archive. The solutions in the archive are ordered according to their quality (determined through objective function evaluations). Each solution has an associated weight proportion to the solution quality. ACOR uses pheromone information to make the probabilistic choice. In the case of robot motion planning and control, the next step for the robot will be determined by the fittest ant (pheromone information) and the robot will move to that step using kinematic equations. The algorithm is divided into the following phases:

### Initialization phase

The algorithm starts with the creation of archive solution during the initialization of ants. ACOR utilizes Gaussian kernal which has three vectors of parameters: weights, means and standard deviations. The time complexity for this phase is O(n), where n is the number of ants.

### Archive population phase

For the archive solution, the means and standard deviations are calculated using Eqs. (1) and (2):

$$\mu = \mu_1, \ldots, \mu_n = s_1, \ldots, s_n,$$ (1)

where $\mu_i$ are the means and $s_i$ the archive solutions for $i = 1 \ldots n$, and

$$sd_i = \xi \sum_{n=1}^{k} \frac{|s_n - s_l|}{k-1},$$ (2)

where $sd_i$ is the standard deviation, $k$ is the archive population, $\xi$ is the convergence speed of the algorithm and $s_l$ is the chosen solution. ACOR stores in archive solution the values of the solutions' n variables and the value of their objective functions. This phase has a time complexity of O(n), where n is the number of ants.

### New population creation phase

This phase involves creation of the new population array which is subsequently randomly initialized. This phase has a time complexity of O(1).

### Solution construction phase

This phase starts which consists of the Gaussian kernel selection and then generating Gaussian random variable. Gaussian kernel selection is based on roulette wheel selection and probability. The probability is computed based on weights. The weights and probability are given in Eqs. (3) and (4):

$$w_i = \frac{1}{qk\sqrt{2*\Pi}} \exp(-0.5\frac{(i-1)^2}{(qk)^2}),$$ (3)

where, $w_i$ is the weight for individual solution, $q$ is the intensification factor and $k$ is the archive population, and

$$P_i = \frac{w_n}{\sum_{n=1}^{k} w_n},$$ (4)

where, $P_i$ is the probability of individual archive solution, $k$ is the archive population and $w_n$ is the weight of individual archive solution.

An ant chooses probabilistically one of the solutions in the archive using Eq. (4).

The position of the new population is the random variable generated using means and standard deviations from the Archive Population Phase and normally distributed random numbers. This phase has a time complexity of O(n), where n is the number of ants.

### Evaluation phase

The new constructed solution is then evaluated using a fitness function and merged with the archive population. Finally, the total population is sorted to get the best solution and a new set of archive solution. The time complexity for this phase is O(1).

All phases except the Initialization phase will be repeated until the robot reaches its destination.

## ROBOT MOTION PLANNING AND CONTROL FORMULATION

In this research, the path planning problem of robots is solved by the ACO while their motion controls by the kinematic equations. For the path planning problem, initial artificial ants and objective functions are used. For motion control, the kinematic equations will be used to control a robot to a point (step location) generated by ants. Kinematic equations which are essentially ODE's governing the motion of robot are used to control the motion of the robot. These equations are dependent on the type of the robot. For example, a point mass robot will have a set of different kinematic equations when compared that of a tractor-trailer robot because the latter may include nonholonomic constraints. This section is further divided into ants representation, multi-objective path planning problem, and the problem formulation of path planning and motion control of robots.

### Multi-objective path planning problem

The robot path planning problem is formulated as the multi-objective problem. The two objectives are obtaining short path and safe path of the robots.

The following is the definition of a point-mass robot adopted from *Prasad et al. (2020a)*:

**Definition 1.** A $j$th point-mass $P_j$ is a disk of radius $rp_j \geq 0$ and is positioned at $(x_j(t), y_j(t)) \in \mathbb{R}^2$ at time $t \geq 0$. Precisely, the point-mass is the set

$$P_j = \{(z_1, z_2) \in \mathbb{R}^2 : (z_1 - x_j)^2 + (z_2 - y_j)^2 \leq rp_j^2\}$$

for $j = 1, 2, \ldots, n$,

**Definition 2.** The target for $P_j$ is a disk of center $(p_{j1}, p_{j2})$ and radius $rt_j$ which is described as

$$T_j = \left\{(z_1, z_2) \in \mathbb{R}^2 : (z_1 - p_{j1})^2 + (z_2 - p_{j2})^2 \leq rt_j^2\right\} \tag{5}$$

for $j = 1, 2, \ldots, n$.

### Ant representation

The motion of each robot will be guided by a colony of ants which are randomly distributed in the working environment. Since we are considering $n$ point-mass robots, there will be $n$ colonies altogether with each colony having $n_j$ ants. The following is the definition of a moving ant in the $j$th colony:

**Definition 3.** The $i$th ant in the $j$th colony $A_{ij}$ is a disk of radius $ra_{ij} \geq 0$ and is positioned at $(xa_{ij}(t), ya_{ij}(t)) \in \mathbb{R}^2$ at time $t \geq 0$. Precisely, the $i$th ant in the $j$th colony is the set

$$A_{ij} = \{(z_1, z_2) \in \mathbb{R}^2 : (z_1 - xa_{ij})^2 + (z_2 - ya_{ij})^2 \leq ra_{ij}^2\}$$

for $i = 1, 2, \ldots, n$ and $j = 1, 2, \ldots, n$.

### Short path

A robot will have a target seeking behaviour, that is, a robot should always be at a minimum distance from the target while it navigates through the cluttered environment. Therefore, the total length of the robot's path should be a minimum. Ants will be used to guide the robot to the target. In case of a multi-robot environment, each robot will have a set of ants associated with it for navigation. Since ants are used to plan the path for the robot, the fittest ant will be chosen for the robot's next step location. For each ant in the $j$th colony, an Euclidean formula shown in Eq. (6) is used to calculate the distance between the $i$ th ant and the target:

$$d_{ij} = \sqrt{(p_{j1} - xa_{ij})^2 + (p_{j2} - ya_{ij})^2}. \tag{6}$$

### Safe path

A path is safe if it has no obstacles in it. However, the workspace considered for this research has multiple obstacles, therefore obstacle avoidance becomes necessary. There are two types of obstacles used in this paper: (1) circular obstacles and (2) line segments. The following are the definitions of a circular obstacle and a line segment, adopted from *Prasad et al. (2020a)*:

**Definition 4.** The $l$th circular obstacle with center $(o_{l1}, o_{l2})$ and radius $ro_l > 0$ on the $z_1z_2$ plane is described as

$$FO_l = \left\{ (z_1, z_2) \in \mathbb{R}^2 : (z_1 - o_{l1})^2 + (z_2 - o_{l2})^2 \leq ro_l^2 \right\},$$

for $l = 1, 2, \ldots, q$.

**Definition 5.** The $k$th line segment in the $z_1z_2$ plane, from the point $(a_{k1}, b_{k1})$ to the point $(a_{k2}, b_{k2})$ is the set

$$LO_k = \left\{ (z_1, z_2) \in \mathbb{R}^2 : (z_1 - a_{k1} - \lambda_k(a_{k2} - a_{k1}))^2 + (z_2 - b_{k1} - \lambda_k(b_{k2} - b_{k1}))^2 = 0 \right\},$$

where $\lambda_k \in [0, 1]$, $k = 1, 2, \ldots, m$.

For a circular obstacle, Euclidean formula shown in Eq. (7) is used to calculate the shortest distance between $i$th ant in the $j$ colony and the $l$th obstacle:

$$d1_{ijl} = \sqrt{(o_{l1} - xa_{ij})^2 + (o_{l2} - ya_{ij})^2} \tag{7}$$

for $i = 1, 2, \ldots, n_j$, $j = 1, 2, \ldots, n$ and $l = 1, 2, \ldots, q$. Since there are many obstacles, the distance between each ant and the obstacles will be calculated. The sum of the distances between $i$th ant in the $j$th colony and obstacles is given by:

$$f1_{ij} = \sum_{l=1}^{q} d1_{ijl}. \tag{8}$$

To avoid a line segment, the distance between an ant and several points on that line segment are calculated, and the point generating minimum distance is considered. This technique is known as *minimum distance technique* (MDT) that has been adopted from *Sharma, Vanualailai & Singh (2014)*. Avoiding the closest point on a line segment at any

given time will result in avoiding the entire line segment. Again, the Euclidean formula is used to calculate the distance of an ant with a point on the line segment:

$$d2_{ijk} = \sqrt{(a_{k1} + \lambda_{ijk}(a_{k2} - a_{k1}) - xa_{ij})^2 + (b_{k1} + \lambda_{ijk}(b_{k2} - b_{k1}) - ya_{ij})^2}, \tag{9}$$

where $\lambda_{ijk} \in [0,1]$, $i = 1, 2, \ldots, n_j$, $j = 1, 2, \ldots, n$ and $k = 1, 2, \ldots, m$. Like circular obstacles, there are many line segments, therefore the distance a point on the line segment and an ant will be calculated for each line segment and will be summed as:

$$f2_{ij} = \sum_{k=1}^{m} d2_{ijk}. \tag{10}$$

In a multi-robot environment, the inter-collisions between robots must be avoided. Since ants determine the next step location of a robot, each ant in the $j$th colony must avoid other robots which is considered as a moving obstacle. The Euclidean distance as shown in Eq. (11) is used to avoid robots from colliding with each other:

$$d3_{ijh} = \sqrt{(xa_{ij} - x_h)^2 + (ya_{ij} - y_h)^2}, \tag{11}$$

for $i = 1, 2, \ldots, n_j$, $j = 1, 2, \ldots, n$, $h = 1, 2, \ldots, n$, $h \neq j$.

The sum of the distances between each ant and different robots is given by:

$$f3_{ij} = \sum_{\substack{h=1 \\ h \neq j}}^{n} d3_{ijh}. \tag{12}$$

## Problem formulation

The problem is the minimization optimization problem which finds the optimal path for mobile robots in a cluttered environment. The fitness equation is designed by summing all the objective functions defined in this section:

$$f_{ij} = a.\frac{1}{f1_{ij}} + b.\frac{1}{f2_{ij}} + c.\frac{1}{f3_{ij}} + d.d_{ij} \tag{13}$$

which is the fitness of $i$th ant for robot $j$ and $a$, $b$, $c$ and $d$ are control parameters. The ant having the minimum $f_{ij}$ will be the fittest ant in the $j$th colony. It means that the ant is located at a safe distance from obstacles and at a minimum distance from the target. The $j$th robot will move to the fittest ant of the $j$th colony and this process will continue until the $j$th robot reaches its target. The control parameters $a$, $b$ and $c$ are the fitting parameters that decide path safety. With a high value of parameter $a$ the ants will avoid the stationary circular obstacles from the greater distance. Similarly, a small value of $b$ will mean that the ants will avoid the line segments from a closer distance which can compromise safety. However, when there is a decrease in the value of $a$, $b$ and $c$, the chances of collision with obstacles, line segments and other robots are high. Likewise, a higher value of $d$ will minimize the path length and a smaller value will maximize the path length. Therefore, a proper selection of these parameters decides the success of the objective function in planning the next step for a robot.

### Motion control

The kinematic equations are used to control the robot to the step location generated ants. The kinematic equations will be used to generate smooth path for the robots in between the nodes.

## PROPOSED ALGORITHM

The proposed new algorithm is a hybrid of the ant colony optimization and kinematic equations, named *ACO-Kinematic*. The choice of ACO variant was made from the results of *Socha & Dorigo (2008a)*; *Socha & Dorigo (2008b)* where they showed that it outperformed other ACO variants and was equivalent to some heuristic algorithms for continuous domain. The robot's next step will be planned using the ant colony optimization algorithm and the robot will move to that step using its kinematic equations. In the literature, there are various hybrid models but according to the authors knowledge, there is none of this kind. The ACO-Kinematic pseudocode is shown in Algorithm 1.

```
Algorithm 1 ACO-Kinematic
Objective function f(x)
Initialize population of ants (Archive size)
Initialize robots initial and target positions
Initialize the weights and selection probabilities
While (robots current position < target position) do
    For all Ants in Archive Size do
        Calculate means
    End For
    For all Ants in Archive Size do
        Calculate standard deviation
    End For
    Create New Population Array (Sample Size)
    For all Ants in Sample Size do
        Construct solution based on Gaussian Kernal
        Evaluate new solutions
    End For
    Merge main population (archive) and new population (sample size)
    Rank ants and find the new best position

    Move the robot from current position to the new best position using
    Kinematic equations
    Update current position to new best
End While
Post-processing the results and visualization;
```

## RESULTS

Table 1 shows the initial parameters of the ACO-Kinematic algorithm used in the three case studies discussed in the following subsections. Parameters 1–8 are used for path planing while parameters 9–10 are used for motion control. Safety parameters *a*, *b* and *c* will be used by users to fine tune avoidance of obstacles, line segments and other robots, respectively. Convergence parameter *d* determines the time it will take for a robot to reach its destination. Faster convergence means compromising the safety of the robot while the slower convergence means that the operation can be costly. Therefore, the parameters in this research have been adjusted with precaution. In this research the authors have deployed the brute force method to generate the values of the control parameters.

### Case study 1: single point-mass robot and multiple obstacles
The proposed algorithm has been used to navigate the point-mass robot from source to destination in a cluttered environment. The kinematic equations governing the motion of a point-mass robot from its initial position $(x_0, y_0)$ to another point $(p_1, p_2)$ are

$$\dot{x} = \alpha_1(p_1 - x), \qquad\qquad \dot{y} = \alpha_2(p_2 - y) \qquad\qquad (14)$$

where $\alpha_1$ and $\alpha_2$ are positive real numbers.

Figures 1 and 2 show two scenarios where the point-mass robot avoids circular and line obstacles to reach its target. The path of the point-mass robot consists of points that have been generated by the ants. The robot moves from one step to another using its kinematic equations.

### Case study 2: multiple point-mass robots and multiple obstacles
The proposed algorithm has been used to plan and control motion of multiple point-mass robots in a multiple obstacles (circular and rectangular shapes) environment. Figure 3 shows the paths of three point-mass robots. The first robot (R1) has a initial position of (5, 45) and the target placed at (45, 5). The second robot (R2) is placed at the initial position (5, 5) and has to reach the target at (45, 45). The initial and target positions for the third robot (R3) are (45, 25) and (5, 25). The three robots start journey from their initial positions and have a goal to achieve, that is, to reach their target positions safely by taking a shortest route moving from one step to another. The three robots avoid all obstacles in their paths and also avoid colliding with each other. Figure 4 shows the first robot (R1) and the second robot (R2) avoiding each other.

### Application: tractor-trailer robot
The proposed algorithm has been used for motion planning and control of a tractor-trailer robot system. We consider a non-standard tractor-trailer robot which comprises of a rear wheel driven car-like vehicle and a hitched two-wheeled passive trailer attached to the rear axel of the vehicle (Fig. 5).

Let $(x, y)$ represent the cartesian coordinates of the tractor robot, $\theta_0$ be its orientation with respect to the *x*-axis, while $\phi$ gives the steering angle with respect to its longitudinal axis. Similarly, let $\theta_1$ denote the orientation of the trailer with respect to the *x*-axis. Letting

**Table 1  ACO-Kinematic parameters.**

| No. | Parameter | Value |
|---|---|---|
| 1 | Population size (Archive) | 300 |
| 2 | Sample size | 10000 |
| 3 | Deviation-distance ratio | 1 |
| 4 | Intensification factor | 0.5 |
| 5 | a | 0.18 |
| 6 | b | 0.18 |
| 7 | c | 0.18 |
| 8 | d | 0.01 |
| 9 | $\alpha_1$ | 0.1 |
| 10 | $\alpha_2$ | 0.1 |

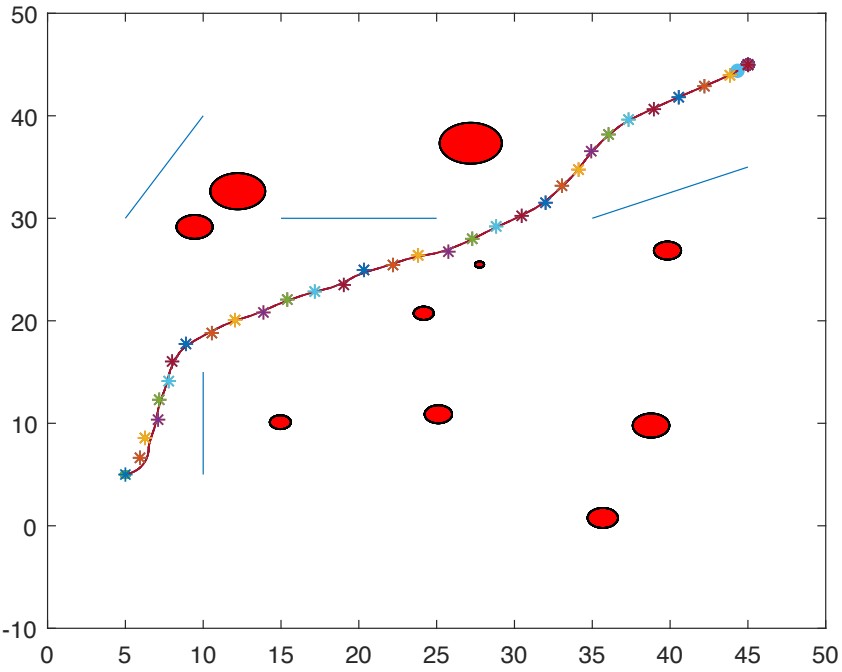

**Figure 1  A point-mass robots path with initial position (5, 5) and target position (45, 45).**

$L$ and $L_t$ be the lengths of the mid-axle of the tractor and trailer, respectively, the motion of the tractor-trailer robot is governed by the following kinematic equations (*Prasad et al., 2020b*)

$$\dot{x} = v\cos\theta_0 - \frac{v}{2}\tan\phi\sin\theta_0,$$
$$\dot{y} = v\sin\theta_0 + \frac{v}{2}\tan\phi\cos\theta_0,$$
$$\dot{\theta}_0 = \frac{v}{L}\tan\phi,$$
$$\dot{\theta}_1 = \frac{v}{Lt}\left(\sin(\theta_0 - \theta_1) - \frac{c}{L}\tan\phi\cos(\theta_0 - \theta_1)\right), \tag{15}$$

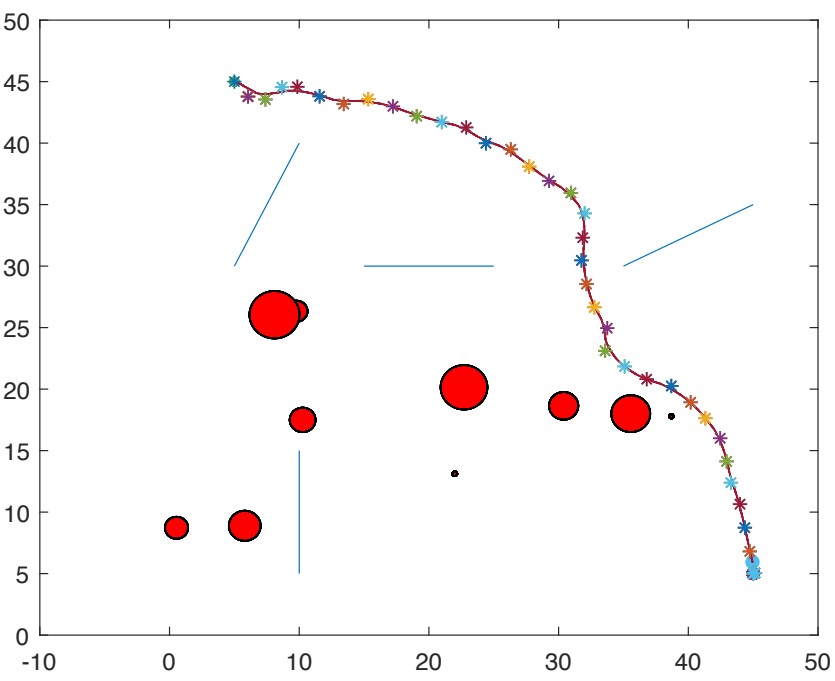

**Figure 2** A point-mass robots path with initial position (5, 45) and target position (45, 5).

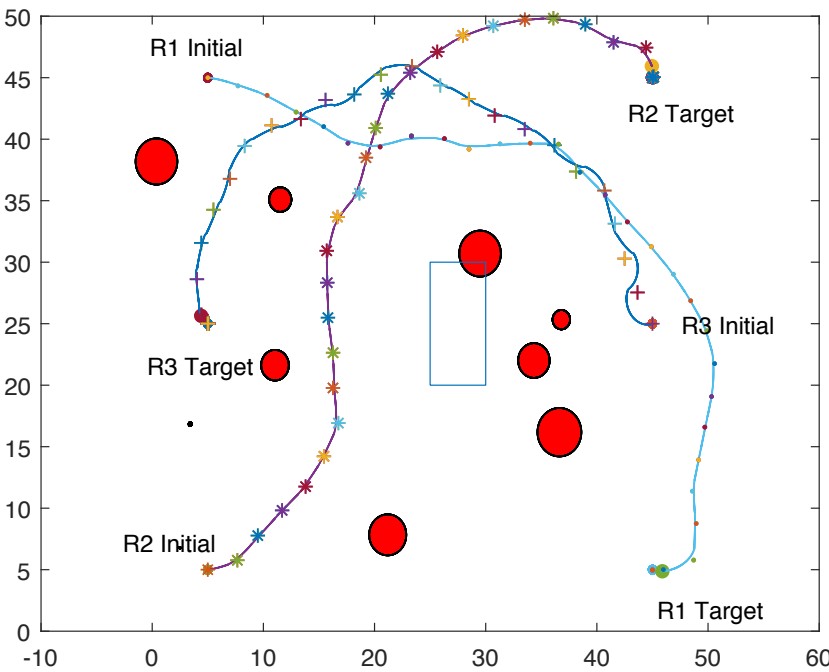

**Figure 3** Paths of three point-mass robots.

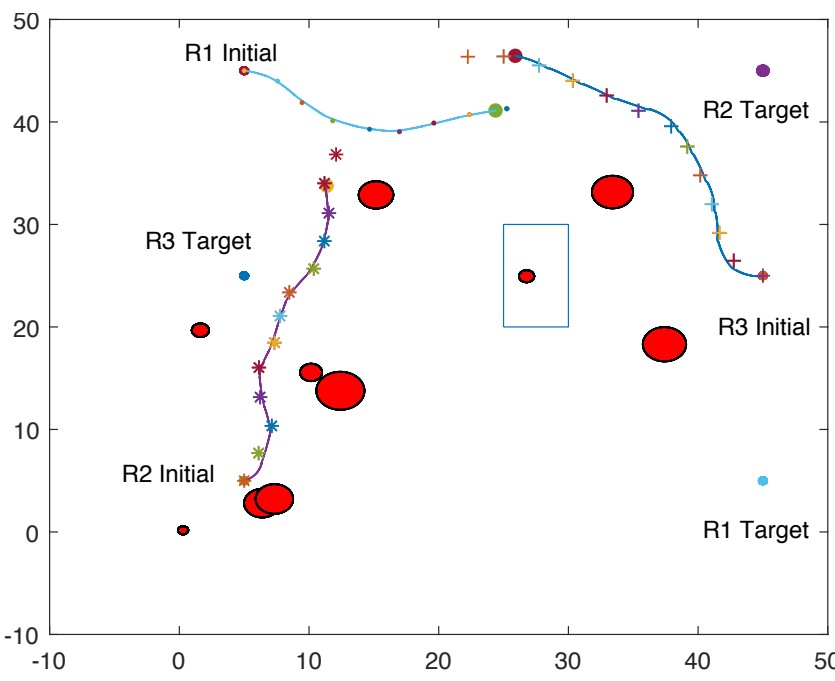

**Figure 4** Partial paths of three point-mass robots with two robots avoiding each other.

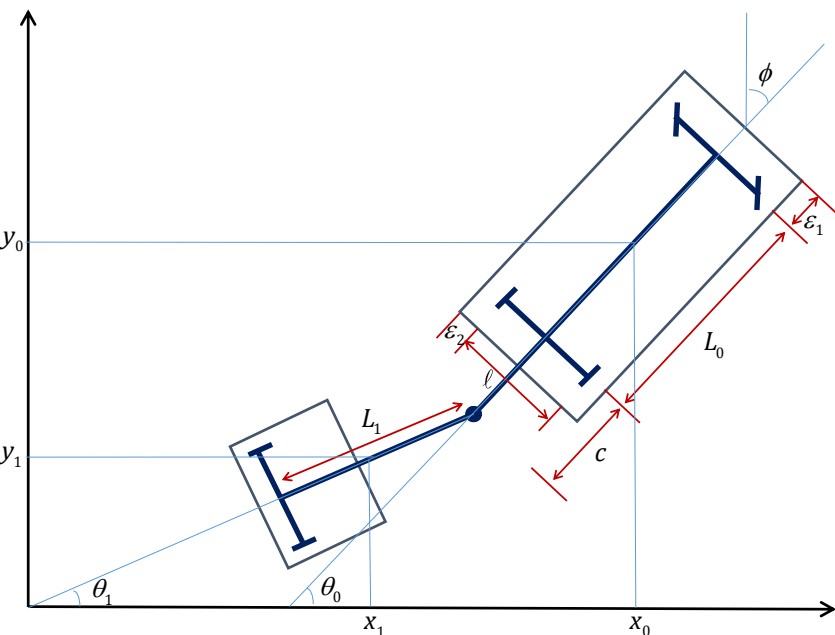

**Figure 5** Schematic representation of the tractor-trailer robot.

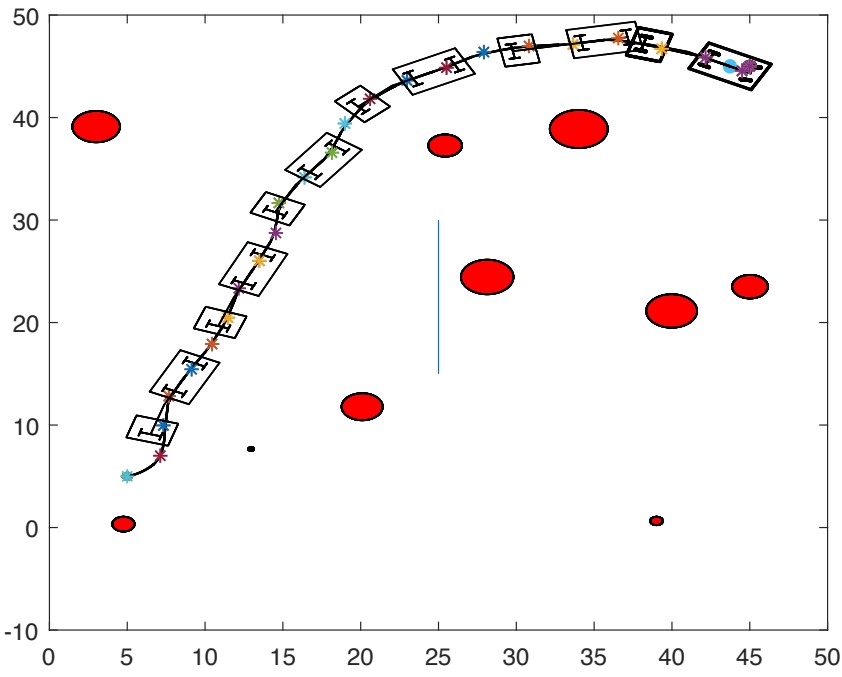

**Figure 6** Path with initial position (5, 5) and target position (45, 45).

where $v$ and $\phi$ which are the translational velocity and the steering angle, respectively, of the tractor robot, given as *Prasad et al. (2020b)*

$$v = \alpha\sqrt{(p_2-y)^2+(p_1-x)^2},$$

$$\phi = \frac{7}{9}\tan^{-1}\left(\xi + \frac{\beta}{\cos|\theta_0-\theta_1|}\right)$$

where $\alpha$ is a positive real number and $\beta = \max\{0, 0.5 - \cos|\theta_1-\theta_0|\} \cdot sign(\theta_1-\theta_0)$. Note that $(p_1, p_2)$ is the next step for the robot generated by ant colony optimization and $(x, y)$ is the current position of the robot. $\xi$ is obtained by numerically solving the differential equation

$$\dot{\xi} = \frac{(p_2-y)\cos\theta_0 - (p_1-x)\sin\theta_0}{\sqrt{(x-p_1)^2+(y-p_2)^2}+0.01} - \text{atan2}(p_2-y, p_1-x) + \theta_0,$$

$$\xi(0) = \text{atan2}(p_2-y(0)), (p_1-x(0)) - \theta_0(0)$$

Figsures 6 and 7 show the trajectories of one tractor-trailer robot in two different scenarios.

Figures 8, 9, 10 and 11 show the paths of the three tractor-trailer robots. Table 2 shows the initial and target positions for each robot.

Figure 8 shows that R1 is avoiding R3 by stopping and waiting for R3 to pass. R1 has successfully avoided R3 and R1 resumes its journey towards the target, as shown in Fig. 9. Figure 10 shows that R2 and R3 avoid each other. The complete paths for the three robots are shown in Fig. 11. The robots avoid each other using the fitness function given in

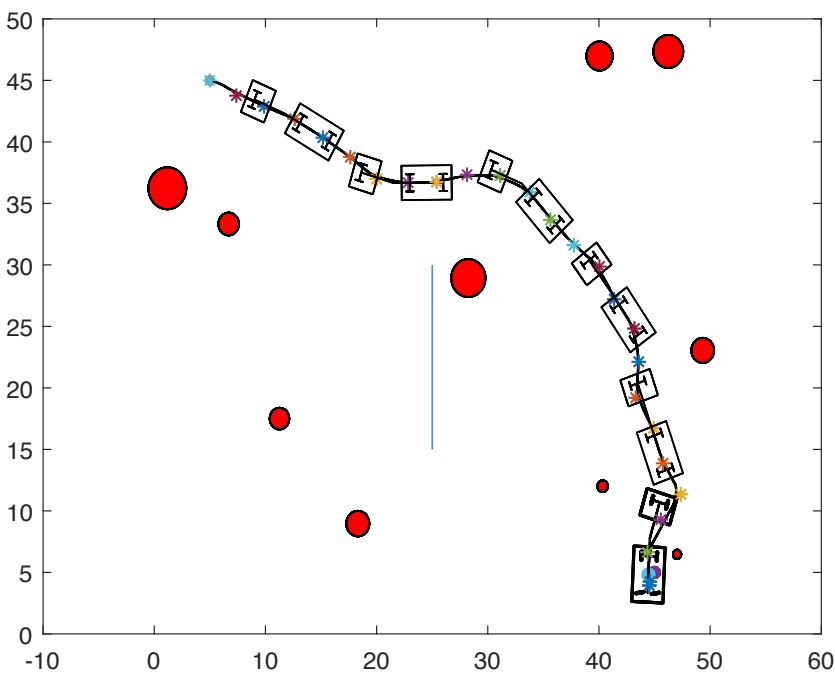

**Figure 7 Path with initial position (5, 45) and target position (45, 5).**

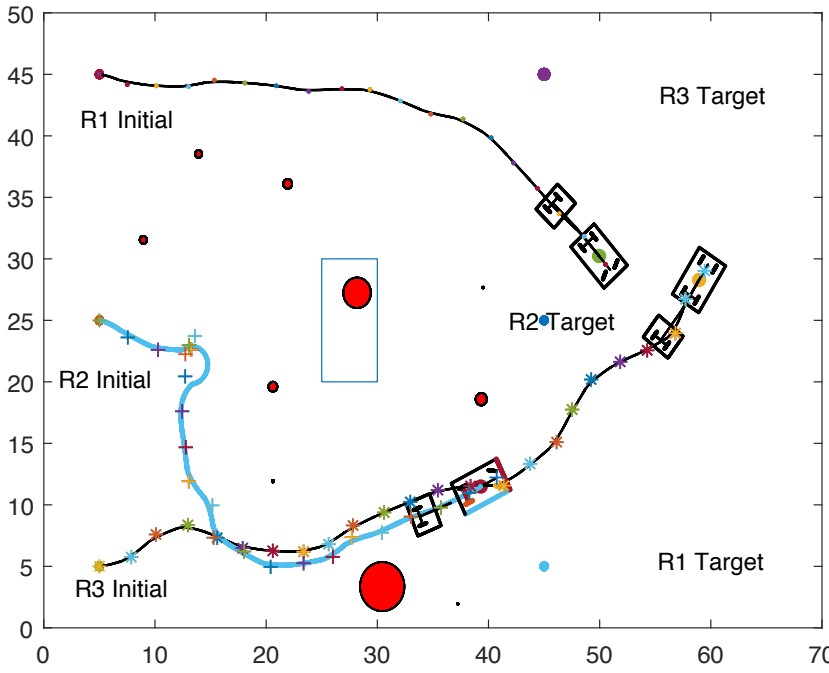

**Figure 8 A tractor-trailer robot (R3) avoiding another tractor-trailer robot (R1).**

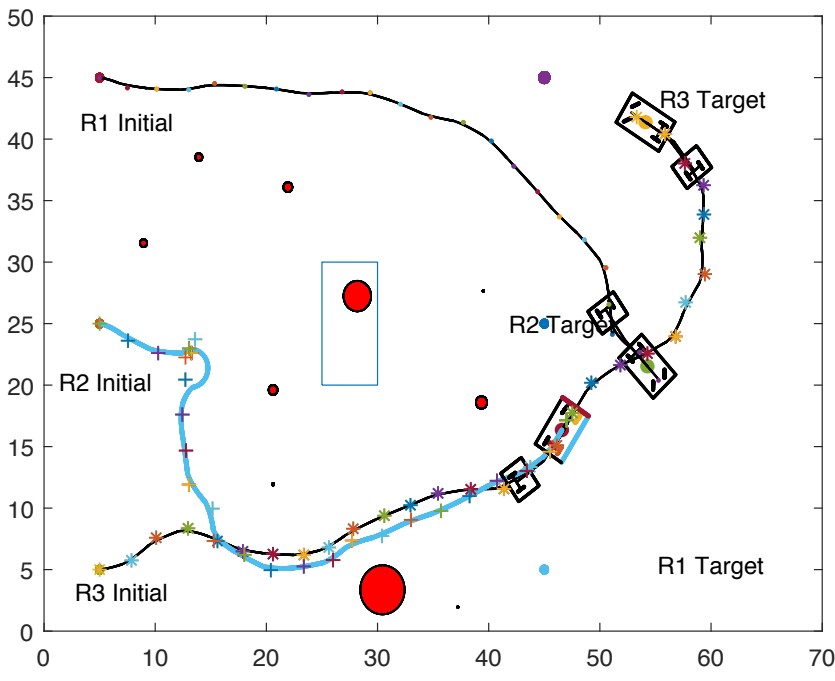

**Figure 9 A tractor-trailer robot (R3) avoiding another tractor-trailer robot (R1).**

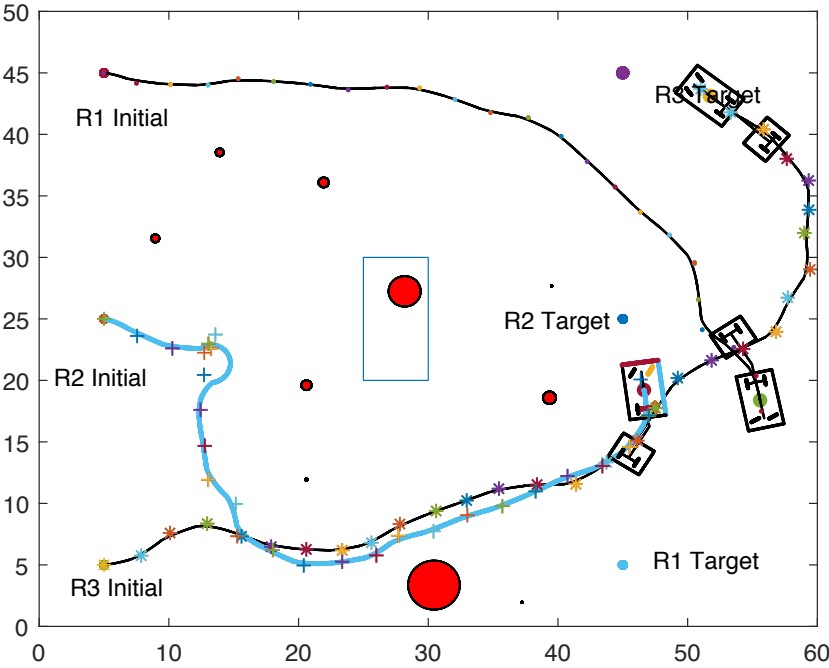

**Figure 10 A tractor-trailer robot (R1) avoiding another tractor-trailer robot (R2).**

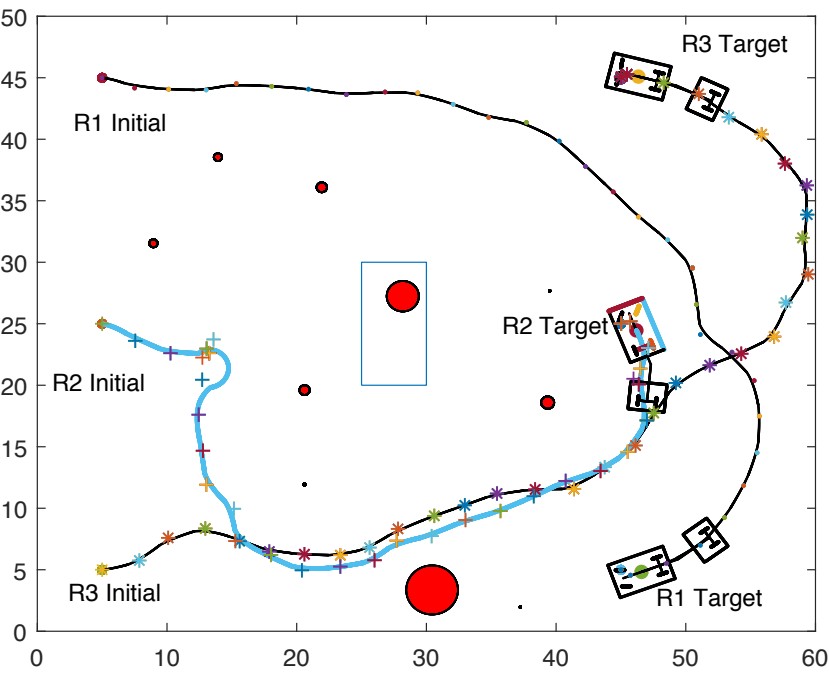

**Figure 11    Complete paths of three tractor-trailer robots.**

**Table 2    Robots' initial and target positions.**

| Robot | Initial position | Target position |
|---|---|---|
| R1 | (5, 45) | (45, 5) |
| R2 | (5, 25) | (45, 25) |
| R3 | (5, 5) | (45, 45) |

Eq. (13) which is used in ACO to determine the fittest ant. The robots are then controlled to the location of the fittest ants by the kinematic equations of the robots.

## DISCUSSION

In this section, the performance of the proposed ACO-Kinematic algorithm will be compared with the Lyapunov-based Control Scheme (LbCS), which is a popular potential field-based method used to solve motion planning and control problem (*Sharma et al., 2018*). The performance will be measured in terms of path length and convergence time. Both algorithms have convergence and safety parameters. For LbCS, a larger convergence parameter value increases convergence time whereas a smaller convergence parameter value will decrease the convergence time. For ACO-Kinematic, a larger convergence parameter value decreases convergence time whereas a smaller convergence parameter value will increase the convergence time. Note that a quicker time to converge can affect a robot's safety. Therefore, the parameters need to be adjusted. There is no method in literature to adjust these parameters apart from the brute-force method. The authors have also used

**Table 3  ACO-Kinematic parameters.**

| Parameter | Point-mass robot | Tractor-trailer | Range | Range source |
|---|---|---|---|---|
| a | 0.02 | 0.02 | 0.01–1 | |
| c | 0.05 | 0.18 | 0.01–1 | |
| d | 0.01 | 0.01 | 0.0001–0.01 | [20] |

**Table 4  LbCS parameters.**

| Parameter | Point-mass robot | Tractor-trailer | Range |
|---|---|---|---|
| beta | 2 | 2 | 0.1–2 |
| beta1 | 0.1 | 0.1 | 0.1–2 |
| beta3 | n/a | 0.2 | 0.1–2 |
| beta4 | n/a | 0.1 | 0.1–2 |
| gamma | n/a | 0.01 | 0.01–2 |
| delta1 | 10 | 10 | 10–20 |
| delta2 | 10 | 10 | 10–20 |

**Table 5  Average path lengths and convergence time of two motion planning and control algorithms.**

| | Algorithms | | | |
|---|---|---|---|---|
| | LbCS | | ACO-Kinematic | |
| Scenario | Time (s) | Path length (cm) | Time (s) | Path length (cm) |
| 1 | 251.02 | 58.57 | 208.72 | 57.98 |
| 2 | 309.44 | 66.3 | 124.99 | 65.35 |
| 3 | 240.92 | 68.34 | 195.79 | 66.27 |

the brute-force method to obtain optimal parameters for the two algorithms that have been used to measure performance as shown in Tables 3 and 4. The LbCS equations and parameters for the point-mass robot and tractor-trailer has been used from *Vanualailai, Ha & Nakagiri (1998)* and *Sharma et al. (2008)*, respectively. The parameter values depend on the type of the robot which is also shown in the two tables. The convergence parameters $d$ for ACO-Kinematic, and *delta*1 and *delta*2 for LbCS have the same values for both robots. Other parameters may differ in their values and are dependent on the type of the robots.

Table 5 shows the average path lengths and the time it takes for a robot to reach the destination using ACO-Kinematic algorithm and LbCS. It shows the results for three scenarios. Robot motion planning and control with ACO-Kinematic algorithm for each scenario was iterated 30 times to calculate the average path length and time. LbCS inherently provides the same results for any run, therefore only 1 run was considered.

For Scenario 1, the trajectories of the point-mass robot for ACO-kinematic and LbCS are shown in Figs. 12 and 13. On average, for ACO-Kinematic algorithm the point-mass robot took 208.72 s to reach the destination with the average path length of 57.98 cm. The best path length was 57.81 cm with the time of 201.4s while the worst path length was 58.29 cm with the time of 210.57s. For LbCS, the point-mass robot took 251.02 s to reach the

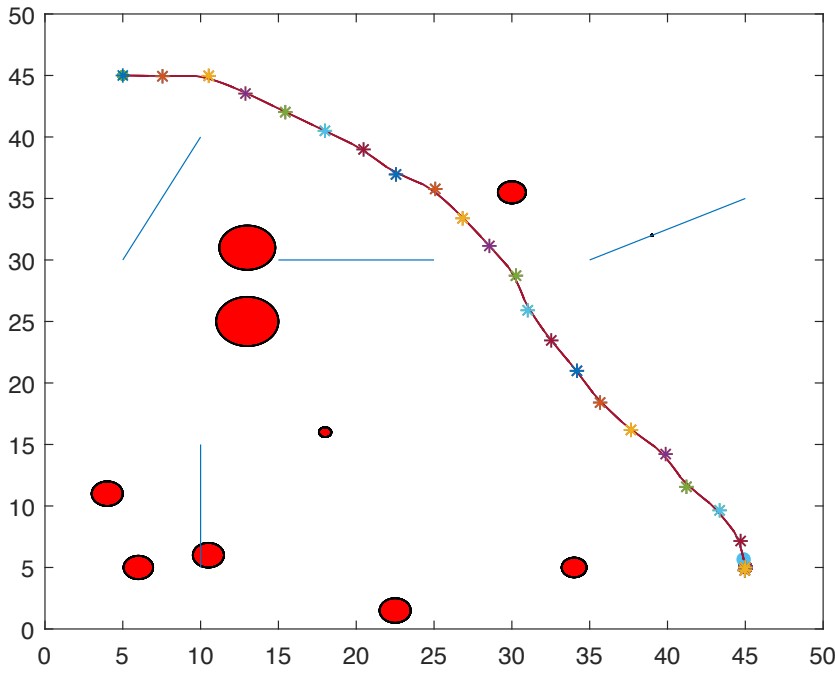

**Figure 12** **Scenario 1: A point-mass robot's path generated by ACO-Kinematic with initial position (5, 45) and target position (45, 5).**

destination with the path length of 58.57 cm. The results of scenario 1 further show that the ACO-Kinematics worst-case path length and time are better than LbCS.

For Scenario 2, the trajectories of the point-mass robot ACO-kinematic and LbCS are shown in Figs. 14 and 15. The point-mass robot with LbCS had a path length of 66.3 cm and took 309.44s to reach the destination while the point-mass robot with ACO-Kinematic took 124.99s to cover the path length 65.35 cm. The best-case path length for ACO-Kinematic was 64.5 cm but it took the robot 138.14s to reach the destination. The worst-case path length for ACO-Kinematic was 66.28 cm and had a time of 101.28s.

For Scenario 3, the point-mass robots was replaced by a non-standard tractor-trailer robotic systems. Figures 16 and 17 show the trajectories of the tractor-trailer robot for ACO-Kinematic and LbCS, respectively. The tractor-trailer robot with the LbCS had a path length of 68.34 cm and time of 240.92s. The robot with ACO-Kinematic had the average path length of 66.27 cm and time of 195.79s. The ACO-Kinematic also provided the best-case path length of 62.99 cm with a time of 230.02s while the worst-case path length was 68.06 cm with a time of 238.05s.

Overall, in all 3 scenarios, ACO-Kinematic was able to achieve the shorter path in a time lesser than that of LbCS.

Figures 18 and 19 show the trajectories of the point-mass robots using ACO-Kinematic and LbCS for Scenario 4. While, the point-mass robot controlled by ACO-Kinematic was able to reach the target, the point-mass robot through LbCS controllers could not. This is because the LbCS system had entered into the local minima, which is one issue that most

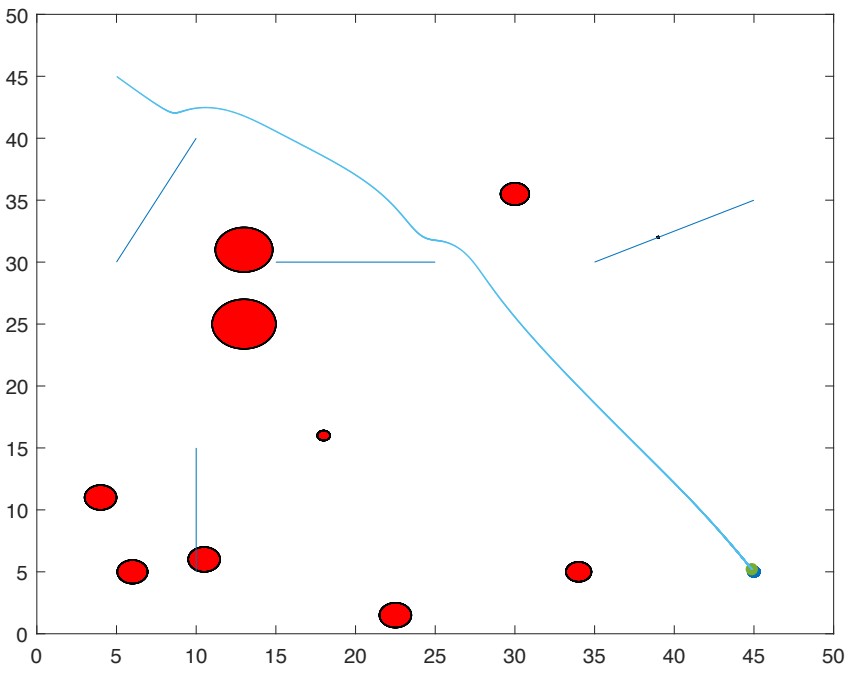

**Figure 13** Scenario 1: A point-mass robot's path generated by LbCS with initial position (5, 45) and target position (45, 5).

artificial potential field methods face. The ACO-Kinematic algorithm was able to solve the problem of local minima and hence a better performer than a traditional motion planning and control algorithms like LbCS.

## CONCLUSION AND FUTURE WORK

In this paper, a unique approach is proposed for solving motion planning and control problems. In particular, a new hybrid algorithm, ACO-Kinematic strategically made up of ant colony optimization and kinematic equations is presented. The new algorithm plans a step for a robot using ant colony optimization and the robot moves to that step using its kinematic equations. The algorithm is inherently capable of making the robot avoid obstacles and other robots while moving from initial to the target position. In the hybrid algorithm, the ACO plans the robot's next step while the kinematic equations controls the robot to that step location. In the authors' belief, this is the first time a hybrid algorithm of this kind is proposed for motion planning and control problem. The algorithm solves a multi-objective problem that consists of finding the safest and shortest path, successfully using kinematic equations to move the robot from one step to another and finally conveys to the final destination.

The ACO-Kinematic algorithm was used in three case studies. In the first case study, a point-mass robot navigated from initial position to target while avoiding multiple circular and line obstacles. The obstacles were static and the locations were known to robots. The second case study extended to multiple point-mass robots in a cluttered environment.

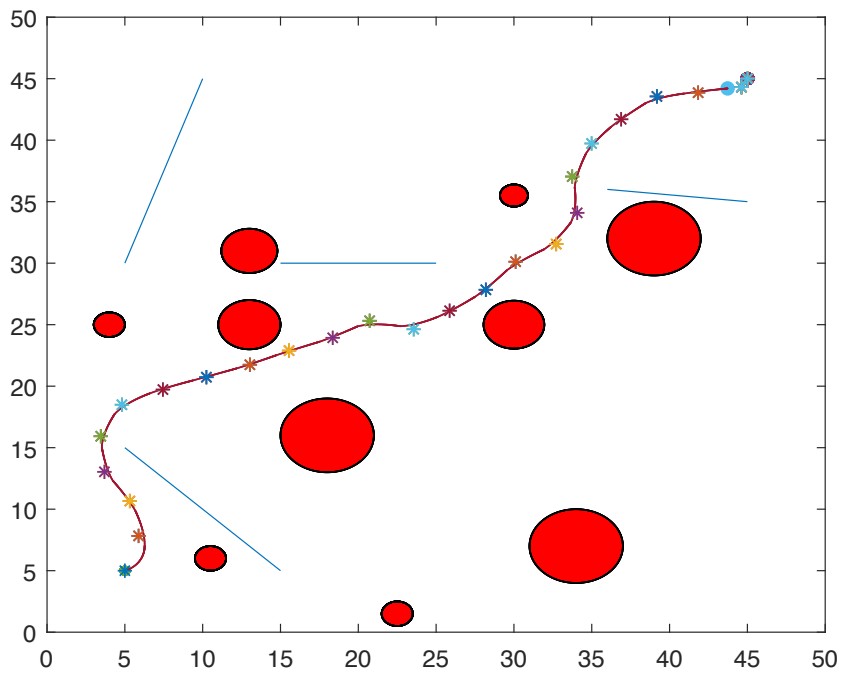

**Figure 14** Scenario 2: A point-mass robot's path generated by ACO-Kinematic with initial position (5, 5) and target position (45, 45).

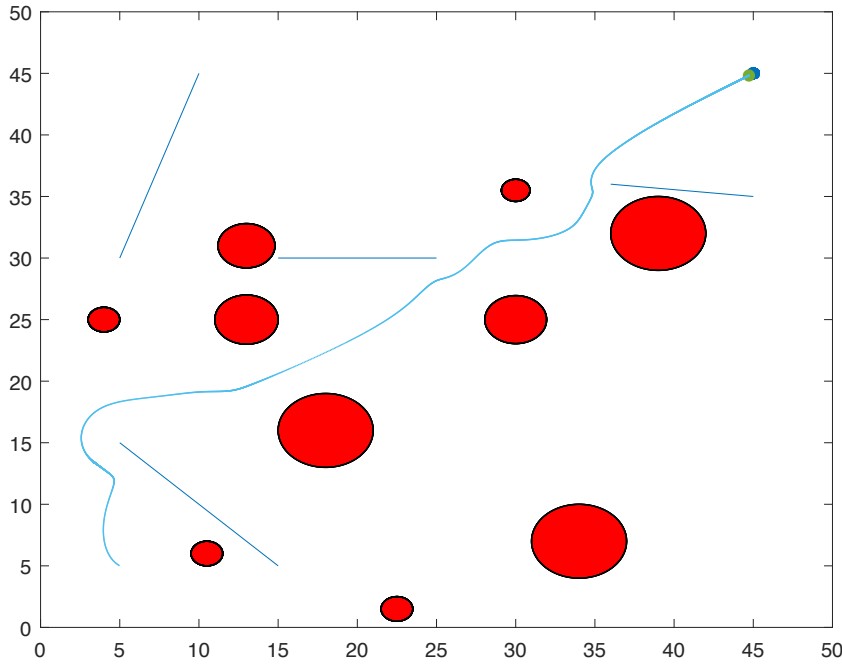

**Figure 15** Scenario 2: A point-mass robot's path generated by LbCS with initial position (5, 5) and target position (45, 45).

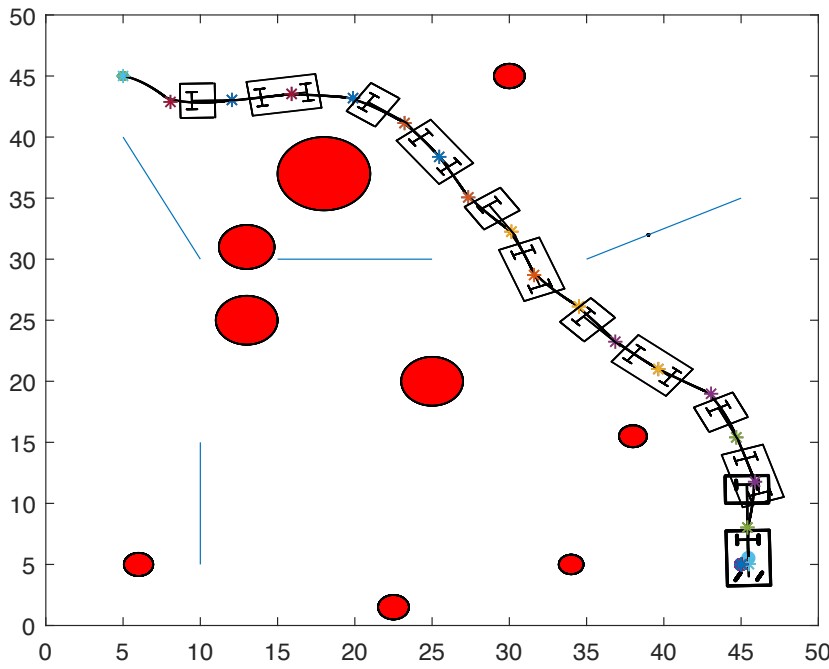

**Figure 16** Scenario 3: A tractor-trailer robot's path generated by ACO-Kinematic with initial position (5, 45) and target position (45, 5).

In addition, the robots also avoided each other. The third case study was an application involving the non-standard 1-trailer robots. The tractor-trailer robots avoided obstacles and other tractor -trailer robots to reach the target from initial position.

The proposed algorithm, was also compared for performance with the Lypanouv based control scheme (LbCS) that has been used in a number of research on motion planning and control problems. Three scenarios were used for both algorithms and path lengths and convergence times were noted. The results showed that ACO-Kinematic algorithm was able to achieve shorter path lengths in relatively shorter convergence times in all three scenarios. It was also noted that LbCS was trapped in the local minima in the fourth scenario and the point-mass robot was not able to reach the target. The point-mass robot controlled by ACO-Kinematic algorithm was able to reach the target in the same scenario. This shows an arterial advantage of the ACO-Kinematic algorithm.

In the future work, the authors will apply ACO-Kinematic to static obstacles with unknown locations, dynamic obstacles and also to other mechanical systems. The authors will also carry out experimental analysis for verifications, comparisons and theoretical proofs of the new algorithm. Since this paper is a first for ACO-Kinematic hybrid , the authors will also upgrade the algorithm by adding new features including heuristic information to be part of ACO selection rule, handle noise in robot control and sensing, and handling imperfect knowledge of the environment.

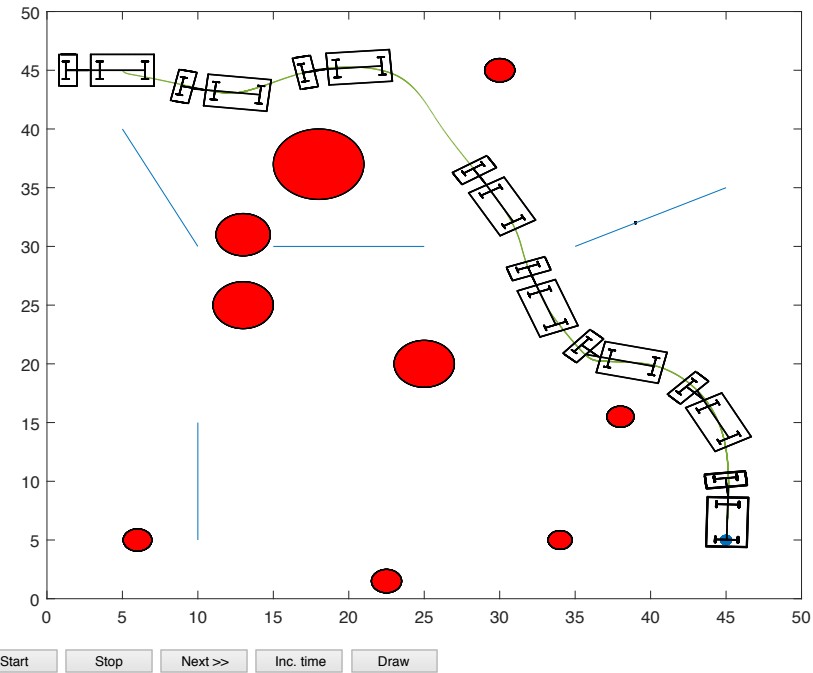

**Figure 17** Scenario 3: A tractor-trailer robot's path generated by LbCS with initial position (5, 45) and target position (45, 45).

### Funding
The authors received no funding for this work.

### Competing Interests
The authors declare there are no competing interests.

### Author Contributions
- Kaylash Chaudhary conceived and designed the experiments, performed the experiments, performed the computation work, prepared figures and/or tables, authored or reviewed drafts of the paper, and approved the final draft.
- Avinesh Prasad performed the computation work, prepared figures and/or tables, authored or reviewed drafts of the paper, and approved the final draft.
- Vishal Chand performed the experiments, prepared figures and/or tables, and approved the final draft.
- Bibhya Sharma performed the experiments, authored or reviewed drafts of the paper, and approved the final draft.

### Data Availability
The MATLAB code used for simulations are available in the Supplementary Files.

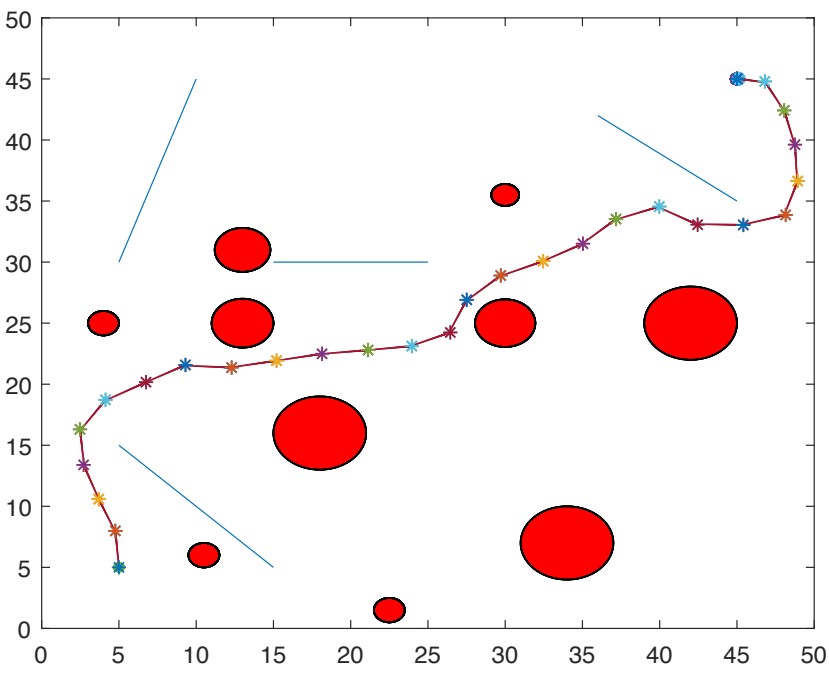

**Figure 18** Scenario 4: A point-mass robot's path generated by ACO-Kinematic with initial position (5, 5) and target position (45, 45).

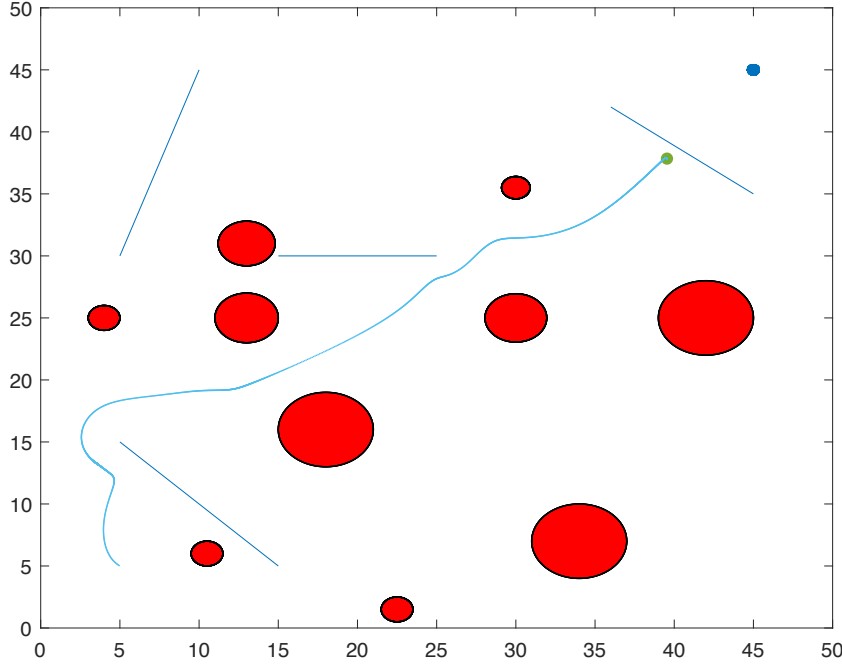

**Figure 19** Scenario 4: A point-mass robot's path generated by LbCS with initial position (5, 5) and target position (45, 45).

## Supplemental Information

Supplemental information for this article can be found online at http://dx.doi.org/10.7717/peerj-cs.905#supplemental-information.

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
