# Peer review of "ACO-Kinematic: a hybrid first off the starting block"

_PeerJ Computer Science, doi:10.7717/peerj-cs.905_

## Round 0.1 · original submission · Major Revisions

Based on the review comments I am suggesting you to submit the revised version of your manuscript.

Reviewer 1 ·

Basic reporting

1. When several titles are cited in the text it is better to be in increasing order (lines 33,53, 103, 114)
2. Use same format, when several titles are cited : [a] [b] [c] or [a,b,c] (line 103)
3. line 134. Section 4 is too short. the section 3 can become "problem statement and objectives"
4. line 140. After mentioning Dorigo include citation of his work.
5. line 141. The first who proposed variant of ACO for continues optimisation is Patrick Siarry (Continuous interacting ant colony algorithm based on dense heterarchy,
J Dréo, P Siarry, Future Generation Computer Systems 20 (5), 841-856, 2004), cite him together with [32]
6. line 169. Remove section 6. Kinematic equations are discussed in section 9.
7. line 207. equation (13). This equation do not show the path length and minimisation of this function do not give the shorter path. The objective function must be the path length with end point the target. The collision avoiding can be constrain. Safety avoiding of obstacles can be solved defining some safety distance from obstacles.

Experimental design

The paper needs new experiments after rewriting the objectives and constraints.

Validity of the findings

The findings will be valid after recalculating with new objective function.

·

Basic reporting

No Comment

Experimental design

No Comment

Validity of the findings

No Comment

Additional comments

This paper presents a combined technique based on ant colony optimization (ACO) algorithm and kinematic equations, named ACO-Kinematic, to solve the robot navigation problem in static environments. In this method, ACO is used to find a collision-free route to the next step, while kinematic equations are used to control and move the robot to the new selected step. The paper can be considered for publication in PeerJ Computer Science, if the following minor and major comments would be carefully addressed:

1) The main limitation of this study is to utilize the ACO-Kinematic algorithm for path planning of the mobile robot in static environments, i.e., considering only static obstacles with known location. Although the authors correctly addressed it as a limitation of their work in Conclusion, there is still a major issue: What about the static obstacles with unknown locations? Is the robot does not aware of the location of obstacles (even static), it cannot run ACO prior to find the full path for the mobile robot.

2) In the original ACO, pheromone and heuristic information (as available) are used to calculate the probability of the different choices (i.e., next steps in your study) using the ACO selection rule. Please provide more details about how next steps are selected via ACO? Why you did not consider heuristic information in your model? For example, the distance to the obstacles and angle to the target can be used as very informative heuristic information not only to speed-up the algorithm, but also to improve the solution quality. There is a fuzzy heuristic based ACO (combining fuzzy heuristic information and pheromone): “FH-ACO: Fuzzy heuristic-based ant colony optimization for joint virtual network function placement and routing”, recently published in Applied Soft Computing, 107, 107401. It utilized multi-criteria fuzzy heuristic as the heuristic information to guide ACO in finding better solutions, and utilizes a multi-criteria fuzzy heuristic model as well as pheromone to construct the full path. You should mention this paper in Introduction or Literature Review, and discuss why you did not consider heuristic information in ACO selection rule?

3) How uncertainties are handled in your model? You should have a plan to handle uncertainties of the environment, or even discuss about it as a limitation of your work in Conclusion. Even for static obstacles, what is your plan if there are uncertainties in the location of the obstacles?

4) Please provide a time complexity analysis for the different phases of the ACO-Kinematic.

5) The results of the ACO-Kinematic should be compared and justified with recently published heuristic- or metaheuristic-based path planning techniques.

6) Finally, the paper should be carefully double-checked to be free of errors.

This manuscript can be reconsidered for the publication in PeerJ Computer Science, if the above issues would be carefully addressed.

---

## Round 0.2 · accepted · Accept

I am happy to inform you that the reviewers are satisfied with the revised manuscript. Thus your manuscript is provisionally accepted for publication.

Reviewer 1 ·

Basic reporting

The authors corrected the paper according reviewers comments

Experimental design

The authors corrected the paper according reviewers comments.

Validity of the findings

The authors corrected the paper according reviewers comments.

Additional comments

The paper is corrected according reviewers comments.

·

Basic reporting

No comment

Experimental design

No comment

Validity of the findings

No comment

Additional comments

The revised version has been efficiently improved, and all of my comments have been corrected. The current version of the manuscript can be accepted for publication.